# CoLES: Contrastive Learning for Event Sequences with Self-Supervision

## Abstract

We address the problem of self-supervised learning on discrete event sequences generated by real-world users. Self-supervised learning incorporates complex information from the raw data in low-dimensional fixed-length vector representations that could be easily applied in various downstream machine learning tasks. In this paper, we propose a new method CoLES, which adopts contrastive learning, previously used for audio and computer vision domains, to the discrete event sequences domain in a self-supervised setting. Unlike most previous studies, we theoretically justify under mild conditions that the augmentation method underlying CoLES provides representative samples of discrete event sequences. We evaluated CoLES on several public datasets and showed that CoLES representations consistently outperform other methods on different downstream tasks.

## 1 Introduction

A promising and rapidly growing approach known as self-supervised learning[1] is the main choice for pre-training in situations where the amount of labeled data for the target task of interest is limited. Most of the research in the area of self-supervised learning has been focused on the core machine learning domains, including NLP (e.g., ELMO (Peters et al., 2018), BERT (Devlin et al., 2019)), speech (e.g., CPC (van den Oord et al., 2018)) and computer vision (Doersch et al., 2015; van den Oord et al., 2018). However, there has been very little research on self-supervised learning in the domain of discrete event sequences, including user behavior sequences (Ni et al., 2018) such as credit card transactions at banks, phone calls and messages at telecom, purchase history at retail and click-stream data of online services. Produced in many business applications, such data is a major key to the growth of modern companies. User behavior sequence is attributed to a person and captures regular and routine actions of a certain type. The analysis of these sequences constitutes an important sub-field of machine learning (Laxman et al., 2008; Wiese and Omlin, 2009; Zhang et al., 2017; Bigon et al., 2019).

NLP, audio and computer vision domains are similar in the sense that the data of this type is "continuous": a short term in NLP can be accurately reconstructed from its context (like a pixel from its neighboring pixels). This fact underlies popular NLP approaches for self-supervision such as BERT's Cloze task (Devlin et al., 2019) and approaches for self-supervision in audio and computer vision, like CPC (van den Oord et al., 2018). In contrast, for many types of event sequence data, a single token cannot be determined using its nearby tokens, because the mutual information between a token and its context is small. For this reason, most state-of-the-art self-supervised methods are not applicable to event sequence data.

In this paper, we propose the *COntrastive Learning for Event Sequences (CoLES)* method that learns low-dimensional representations of discrete event sequences. It is based on a novel theoretically grounded data augmentation strategy, which adapts the ideas of contrastive learning (Xing et al., 2002; Hadsell et al., 2006) to the discrete event sequences domain in a self-supervised setting. The aim of contrastive learning is to represent semantically similar objects (*positive pairs* of images, video, audio, etc.) closer to each other, while dissimilar ones (*negative pairs*) further away. Positive pairs are obtained for training either *explicitly*, e.g., in a manual labeling process or *implicitly* using different data augmentation strategies (Falcon and Cho (2020)). We treat explicit cases as a

---

[1]See, e.g., keynote by Yann LeCun at ICLR-20: https://www.iclr.cc/virtual_2020/speaker_7.html

*supervised* approach and implicit cases as a *self-supervised* one. In most applications, where each person is represented by one sequence of events, there are no explicit positive pairs, and thus only self-supervised approaches are applicable. Our CoLES method is self-supervised and based on the observation that event sequences usually possess periodicity and repeatability of their events. We propose and theoretically justify a new augmentation algorithm, which generates sub-sequences of an observed event sequence and uses them as different high-dimensional views of the same (sequence) object for contrastive learning.

Representations produced by the CoLES model can be used directly as a fixed vector of features in some supervised downstream task (e. g. classification task) similarly to (Mikolov et al., 2013; Song et al., 2017; Zhai et al., 2019). Alternatively, the trained CoLES model can be fine-tuned (Devlin et al., 2019) for the specific downstream task. We applied CoLES to several user behavior sequence datasets with different downstream classification tasks. When used directly as feature vectors, CoLES representations achieve strong performance comparable to the hand-crafted features produced by data scientists. We demonstrate that fine-tuned CoLES representations consistently outperform methods based on other representations by a significant margin. We provide the full source code for all the experiments described in the paper[2].

This paper makes the following contributions: (1) We present the CoLES method that adapts contrastive learning in the self-supervised setting to the discrete event sequence domain. (2) We propose a novel theoretically grounded augmentation method for discrete event sequences. (3) We demonstrate that CoLES consistently outperforms previously introduced supervised, self-supervised and semi-supervised learning baselines adapted to the event sequence domain. We also conducted a pilot study on event sequence data of a large European bank. We tested CoLES against the baselines and achieved superior performance on downstream tasks which produced significant financial gains, measured in hundreds of millions of dollars yearly.

The rest of the paper is organized as follows. In the next section, we discuss related studies on self-supervised and contrastive learning. In Section 3 we introduce our new method CoLES for discrete event sequences. In Section 4 we demonstrate that CoLES outperforms several strong baselines including previously proposed contrastive learning methods adapted to event sequence datasets. Section 5 is dedicated to the discussion of our results and conclusions.

## 2 RELATED WORK

Contrastive learning has been successfully applied to constructing low-dimensional representations (embeddings) of various objects, such as images (Chopra et al., 2005; Schroff et al., 2015), texts (Reimers and Gurevych, 2019), and audio recordings (Wan et al., 2018). The aim of these studies is to identify the object based on its sample (Schroff et al., 2015; Hu et al., 2014; Wan et al., 2018). Therefore, their training datasets explicitly contain several independent samples per each particular object, which form positive pairs as a critical component for learning. These supervised approaches are not applicable to our setting.

For situations when positive pairs are not available or their amount is limited, augmentation techniques were proposed in the computer vision domain. One of the first frameworks with augmentation was proposed by Dosovitskiy et al. (2014). In this work, surrogate classes for model training were introduced using augmentations of the same image. Several recent works (Bachman et al., 2019; He et al., 2019; Chen et al., 2020) extended this idea by applying contrastive learning methods, they are nicely summarised by Falcon and Cho (2020). Although augmentation techniques proposed in these studies provide good performance empirically, we note that no theoretical background behind different augmentation approaches has been proposed so far.

Contrastive Predictive Coding (CPC) is a self-supervised learning approach proposed for non-discrete sequential data (van den Oord et al., 2018). CPC extracts meaningful representations by predicting latent representations of future observations of the input sequence and using autoregressive methods. CPC representations demonstrated strong performance on four distinct domains: audio, computer vision, natural language and reinforcement learning. We adapted the CPC based approach to the domain of discrete event sequences and compared it with our CoLES approach (see Section 4.2).

---

[2]https://github.com/***/*** (the link was anonymized for the double-blind peer review purposes)

Independently of our study, several papers appeared in the past few months on self-supervision for user behavior sequences in the recommender systems domain. Zhou et al. (2020a) proposed to use a CPC-like approach for self-supervised learning on user clicks history. Ma et al. (2020) used an auxiliary self-supervised loss on click sequences. Zhou et al. (2020b) proposed "Cloze" task from BERT (Devlin et al., 2019) for self-supervision on purchase sequences. Finally, Yao et al. (2020) adapts a SimCLR-like approach for text-based tasks and tabular data. Although there has been significant progress on contrastive learning, augmentation for contrastive learning does not have any theoretical grounding and is understudied in the domain of discrete event sequences.

## 3  Problem formulation and overview of the CoLES method

### 3.1  Problem formulation

While the method proposed in this paper could be studied in different domains, in this paper we focus on discrete sequences of events. Assume there are some entities $e$, and the life activity of each entity $e$ is observed as a sequence of events $x_e := \{x_e(t)\}_{t=1}^{T_e}$. Entities could be people or organizations or some other abstractions. Events $x_e(t)$ may have any nature and structure (e.g., transactions of a client, click logs of a user), and their components may contain numerical, categorical, and textual fields (see datasets description in Section 4).

According to theoretical framework of contrastive learning proposed in Saunshi et al. (2019), each entity $e$ is a latent class, which is associated with a distribution $P_e$ over its possible samples (event sequences). However, unlike the problem setting of Saunshi et al. (2019), we have no positive pairs, i.e. pairs of event sequences representing the same entity $e$. Instead, we have only one sequence $x_e$ per entity $e$. Formally, each entity $e$ is associated with a latent stochastic process $\{X_e(t)\}_{t=1}^{T_e}$, and we observe *only one* realisation $\{x_e(t)\}_{t=1}^{T_e}$ generated by the process $\{X_e(t)\}$. Our goal is to learn an *encoder $M$* that maps event sequences into a feature space $\mathbb{R}^d$ in such a way that the obtained *embedding* $c_e = M(\{x_e\}) \in \mathbb{R}^d$ of sequence $\{x_e(t)\}_{t=1}^{T_e}$ encodes essential properties of $e$ and disregards any randomness and noise contained in the sequence. That is, embeddings $M(\{x_1\})$ and $M(\{x_2\})$ should be close to each other, if $x_1$ and $x_2$ are sequences generated by the same process $\{X_e(t)\}$, and they should be further away, when generated by distinct processes. The quality of representations can be examined by downstream tasks in the two ways: (1) $c_e$ can be used as a feature vector for a task–specific model, and (2) encoder $M$ can also be (jointly) fine-tuned (Yosinski et al., 2014).

### 3.2  Sampling of surrogate sequences as an augmentation procedure

While we have no access to the latent processes $\{X_e(t)\}$, we need to use augmentation. Most augmentation techniques proposed earlier for continuous domains (such as image jitter, color jitter or random gray scale in computer vision, see Falcon and Cho (2020)) are not applicable to discrete events. A possible approach for augmentation is generating *sub-sequences* of the same event sequence $\{x_e(t)\}$. The idea proposed below resembles the bootstrap method (Efron and Tibshirani, 1994), which enables to generate several bootstrap samples using only one sample of independent datapoints of a latent distribution. However, our setting is different, since we have no independent observations, so we should rely on different data assumptions. The key property of event sequences that represent life activity is periodicity and repeatability of its events (see Figure 4 in the Appendix D for the empyrical observations of these properties for the considered datasets). This is a motivation for the *Random slices* sampling method applied in CoLES, as presented in Algorithm 1. Each sub-sequence is generated from the initial sequence as its connected segment ("slice") using the following three steps. First, the length of the slice is chosen uniformly from possible values. Second, its starting position is uniformly chosen from all possible values. Third, too short (and optionally too long) sub-sequences are discarded. It could seem that the mean length of obtained sub-sequences are less than the mean length of sequences in the dataset. However, we show in the next section that the distribution of sub-sequences is close to the initial distribution in some realistic assumptions. The overview of the CoLES method is presented in Figure 1.

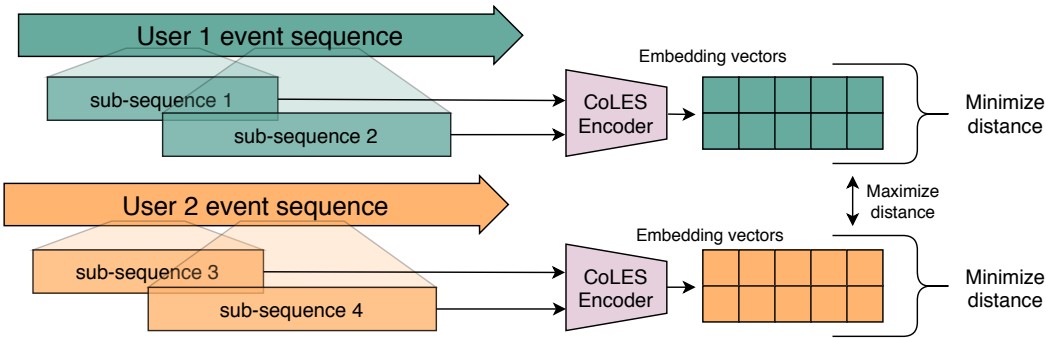

Figure 1: General framework

---

**Algorithm 1:** Random slices sub-sequence generation strategy

---

**hyperparameters:** $m, M$: minimal and maximal possible length of a sub-sequence
$k$: number of trials.
**input:** A sequence $S$ of length $T$.
**output:** $\mathcal{S}$: sub-sequences of $S$.

**for** $i \leftarrow 1$ **to** $k$ **do**
    Generate random integer $T_i$ uniformly from $[1, T]$;
    **if** $T_i \in [m, M]$ **then**
        Generate random integer $s$ from $[0, T - T_i - 1]$;
        Add $S_i := S[s : s + T_i - 1]$ to $\mathcal{S}$
**end**

---

### 3.3 THEORETICAL ANALYSIS

Assume that process $\{X_e(t)\}_{t=1}^{T_e}$ is a segment of a latent process $\{\widehat{X}_e(t)\}_{t=1}^{\infty}$, which generates sequence of all events in the potentially infinite life of entity $e$. That is, we assume that $X_e(t) = \widehat{X}_e(t + s_e)$ for some random starting point $s_e \in \{0, 1, \ldots\}$ and horizon $T_e$. Thus we observe, in our data, segment $[s_e + 1, s_e + T_e]$ of the life of $e$. We also make the following Assumptions:

1. Process $\{\widehat{X}_e(t)\}_{t=1}^{\infty}$ is cyclostationary (in the strict sense) (Gardner et al., 2006) with some period $\widehat{T}$.

2. Starting $s_e$ is independent, and the distribution of $(s_e \mod \widehat{T})$ is uniform over $[0, \widehat{T} - 1]$.

3. Horizon $T_e$ is independent and follows a power–law distribution on $[m, \infty]$.

These assumptions correspond to a scenario where some persons become clients of a service at a random moment and for some random time span and their behaviour obey some periodicity.

**Theorem 1.** *If sequences $\{x_e(t)\}$ in the dataset are generated from latent processes $\{\widehat{X}_e(t)\}$ as described above with a lower bound $m$ for the length of a sequence $\{x_e(t)\}$, then sub-sequences obtained by Algorithm 1 from $\{x_e(t)\}$ follow the same distribution as $\{x_e(t)\}$ up to a slight alteration of the distribution of the length $T_e$. Namely, if $T_e$ follows power law with an exponent $\alpha < -1$, then the density function for the length $T'_e$ of a sub-sequence satisfy*

$$\left(\frac{m-1}{m}\right)^{-\alpha} p(T_e = k) \leq p(T'_e = k) \leq \left(\frac{k}{k - 1/2}\right)^{-\alpha} p(T_e = k) \quad \text{for any } k \in [m, \infty]. \quad (1)$$

This theorem means that a sub-sequence obtained by Algorithm 1 is a representative sample of entity $e$ and follows its latent distribution $P_e$. Combining this result with generalization guarantees proved for setting with explicitly observed positive pairs (Saunshi et al. (2019)), we obtain theoretical background for our implicit self-supervised setting. See Appendix C for the proof of Theorem 1.

### 3.4 MODEL TRAINING

**Batch generation.** The following procedure creates a batch during CoLES training. $N$ initial sequences are randomly taken and $K$ sub-sequences are produced for each of them. Pairs of sub-sequences of the same sequence are used as positive samples and pairs from different sequences are used as negative ones.

We consider several baseline empirical strategies for the sub-sequence generation to compare with Algorithm 1. The simplest strategy is random sampling without replacement. One more strategy is to produce sub-sequences by the random splitting of the initial sequence to several connected segments without intersection between them (see Appendix A).

**Contrastive loss** We consider a classical variant of the contrastive loss, proposed by (Hadsell et al., 2006): $\mathcal{L} = (1 - Y)\frac{1}{2}(D_W^i)^2 + Y * \frac{1}{2}\{max(0, m - D_W^i)\}^2$, where $D_W^i$ is a distance function between embeddings in i-th labeled sample pair, $Y$ is a binary variable identifying that the pair is positive. As proposed in (Hadsell et al., 2006), we use euclidean distance function: $D_W^i = D(A, B) = \sqrt{\sum_i (A_i - B_i)^2}$.

**Pair distance calculation.** In order to select negative samples, we need to compute the pairwise distance between all possible pairs of embedding vectors of a batch. For the purpose of making this procedure more computationally effective we perform normalization of the embedding vectors, i.e. project them onto a hyper-sphere of the unit radius (see Appendix B).

**Negative sampling** is a way to address the following challenge of the contrastive learning approach: using all pairs of samples can be inefficient: for example, some of the negative pairs are already distant enough, thus these pairs are not valuable for the training (Simo-Serra et al., 2015; Schroff et al., 2015). Hence, only a part of possible negative pairs in the batch are used during loss calculation. We compared the most popular choices for negative sampling applied for CoLES, see Section 4.2 for details.

### 3.5 ENCODER ARCHITECTURE

To embed a sequence of events to the fixed-size vector, we use an encoder network, which consists of two conceptual parts: the event encoder and the sequence encoder subnetworks.

**The event encoder** $e$ takes the set of attributes of each single event $x_t$ and outputs its representation in the latent space $\mathbb{R}^d$: $z_t = e(x_t)$. The event encoder consists of several embedding layers and batch normalization layers. Each categorical attribute is encoded by its corresponding embedding layer. Batch normalization is applied to numerical attributes of events. Outputs of all embedding and batch normalization layers are concatenated to produce latent representation $z_t$.

**The sequence encoder** $s$ takes latent representations of the sequence of events: $z_{1:T} = z_1, z_2, \cdots z_T$ and outputs the representation of the whole sequence $c_t$ in the time-step $t$: $c_t = s(z_{1:t})$. Several approaches can be used to encode a sequence (Cho et al., 2014; Vaswani et al., 2017) . In our experiments we use the recurrent network (RNN) similarly to (Sutskever et al., 2014). The output produced for the last event is used to represent the whole sequence of events. In the case of RNN the last output $h_t$ is a representation of the sequence.

To summarise, the CoLES method consists of three major ingredients: event sequence encoder, positive and negative pair generation strategy and the loss function for contrastive learning.

## 4 EXPERIMENTS

We compare our method with existing baselines on several publicly available datasets from various data science competitions. We chose datasets with sufficient amounts of discrete events per user.

**Age group prediction competition**[3]. The dataset of 44M anonymized credit card transactions representing 50k persons was used to predict the age group of a person. The label is known for 30k persons, other 20k are unlabelled. The group ratio is balanced in the dataset. Each transaction includes the date, type, and amount being charged.

---

[3]https://ods.ai/competitions/sberbank-sirius-lesson

**Churn prediction competition**[4]. The dataset of 1M anonymized card transactions representing 10K clients was used to predict a churn probability. Each transaction is characterized by date, type, amount and Merchant Category Code. 5k clients have labels, 5.2k clients haven't labels. Target is binary, almost balanced with proportions 0.55 and 0.45.

**Assessment prediction competition**[5]. The task is to predict the in-game assessment results based on the history of children's gameplay data. Target is one of 4 grades, with proportions 0.50, 0.24, 0.14, 0.12. The dataset consists of 12M gameplay events combined in 330k gameplays representing 18k children. 17.7k gameplays are labeled, the remaining 312k gameplays are not labeled. Each gameplay event is characterized by timestamp, event code, the incremental counter of events within a game session, time since the start of the game session, etc.

**Retail purchase history age group prediction**[6]. The task is to predict the age group of a client based on its retail purchase history. The group ratio is balanced in the dataset. Only labeled data is used. The dataset consists of 45,8M retail purchases representing 400k clients. Each purchase is characterized by time, product level, segment, amount, value, loyalty program points received.

As we can see in Figure 3 (Appendix D), these datasets satisfy the power law assumption for the sequence length distribution of Theorem 1. Also, as shown in Figure 4 (Appendix D) the datasets satisfy the periodicity and repeatability assumption.

**Dataset split.** For each dataset, we set apart 10% persons from the labeled part of the data as the *test set* that we used for evaluation of different models. The rest 90% of labeled data and unlabeled data constitute our *training set* used for learning. For all methods, a random search on 5-fold cross-validation over the training set is used for hyper-parameter selection. The hyper-parameters with the best out-of-fold performance are then chosen. For the learning of semi-supervised/self-supervised techniques (including CoLES), we used all transactions of training sets including unlabeled data. The unlabelled parts of the datasets were ignored while training supervised models.

**Performance.** Neural network training was performed on a single Tesla P-100 GPU card. For the training part of CoLES, the single training batch is processed in 142 milliseconds. For example, in the age group prediction dataset the single training batch contains 64 unique persons with 5 sub-sequences per person, i.e. 320 training sub-sequences in total, the mean number of transactions in a sub-sequence is 90, hence each batch contains about 28800 transactions.

**Hyperparameters** Unless we explicitly specify, we use contrastive loss and random slices pair generation strategy for CoLES in our experiments (see Section 4.2 for motivation). The final set of hyper-parameters used for CoLES is shown in the Appendix E, Table 5.

## 4.1 BASELINES

**LightGBM.** We consider the Gradient Boosting Machine (GBM) method (Friedman, 2001) on hand-crafted features. GBM can be considered as a strong baseline in cases of tabular data with heterogeneous features. (Wu et al., 2009; Vorobev et al., 2019; Zhang and Haghani, 2015; Niu et al., 2019). GBM based model requires a large number of hand-crafted aggregate features produced from the raw transactional data. An example of an aggregate feature is an average spending amount in some categories of merchants, such as hotels of the entire transaction history. We used LightGBM (Ke et al., 2017) implementation of the GBM algorithm with nearly 1,000 hand-crafted features for the application. The details of producing hand-crafted features can be found in the Appendix E.1.

**Self-supervised baselines.**

**NSP.** We consider a simple baseline inspired by the *next sentence prediction* task used in BERT (Devlin et al., 2019). Specifically, we generate two sub-sequences A and B, in a way that 50% of the time B is the sub-sequence from the same sequence as A and follows it (positive pair), and 50% of the time it is a random sub-sequence taken from another sequence (negative pair).

---

[4]https://boosters.pro/championship/rosbank1/
[5]https://www.kaggle.com/c/data-science-bowl-2019
[6]https://ods.ai/competitions/x5-retailhero-uplift-modeling

Table 1: Comparison of batch generation strategies

| Dataset | Random samples | Random disjoint samples | Random slices |
|---|---|---|---|
| **Age group** (Accuracy) | $0.613 \pm 0.006$ | $0.619 \pm 0.011$ | $\mathbf{0.639} \pm 0.006$ |
| **Churn** (AUROC) | $0.820 \pm 0.014$ | $0.819 \pm 0.011$ | $\mathbf{0.823} \pm 0.017$ |
| **Assessment** (Accuracy) | $0.563 \pm 0.004$ | $0.563 \pm 0.004$ | $\mathbf{0.618} \pm 0.009$ |
| **Retail** (Accuracy) | $0.523 \pm 0.001$ | $0.505 \pm 0.002$ | $\mathbf{0.542} \pm 0.002$ |

5-fold cross-validation metric $\pm 95\%$ is shown

**SOP.** Another simple baseline is the same as *sequence order prediction* task from ALBERT (Lan et al., 2020). It uses two consecutive sub-sequences as a positive pair, and two consecutive sub-sequences with swapped order as a negative pair.

**RTD.** We also adapt the *replaced token detection* approach from ELECTRA (Clark et al., 2020) for event sequences as a baseline for our research. We replaced 15% of events from the sequence with random events, taken from other sequences and train a model to predict whether an event is replaced or not.

**CPC.** As the last self-supervised baseline, we selected the recently proposed Contrastive Predictive Coding (CPC) (van den Oord et al., 2018), a self-supervised learning method that produced an excellent performance on sequential data of such traditional domains as audio, computer vision, reinforcement learning and recommender systems (Zhou et al., 2020a).

**Supervised learning**. In addition to the aforementioned baselines, we compare our method with a supervised learning approach where the encoder network $e$ (see Section 3.5) and the classification sub-network $h$ are jointly trained on the downstream task target, i. e. the classification sub-network takes encoder output and produces a prediction: $\hat{y} = h(e(x))$. One-layer neural net with softmax activation is used as $h$. Note that no pre-training is used in this case.

Note that all neural network baselines use the same architecture of the encoder model as CoLES.

## 4.2 RESULTS

**Features of CoLES.** To evaluate the proposed method of sub-sequence generation we compared it with two alternative strategies described in Section 3.4. The results are presented in Table 1. The proposed random slices sub-sequence generation strategy significantly outperforms alternative strategies, what confirm theoretical results (see Section 3.3). Also, note that the random samples strategy is similar to the augmentation strategy proposed by Yao et al. (2020), and the random disjoint samples strategy is similar to sub-sequence generation proposed by Ma et al. (2020).

We evaluated several possible loss functions and found that contrastive loss that can be considered as the basic variant of contrastive learning loss, performs on par or better than other losses on the downstream tasks (see Appendix F.1, Table 7). This means that improvements obtained by more recent losses in object recognition tasks does not necessarily lead to gains in other downstream tasks.

We also compared popular negative sampling strategies (distance-weighted sampling (Manmatha et al., 2017), and hard-negative mining (Schroff et al., 2015)) with random negative sampling strategy. The results are shown in the Appendix F.1, Table 8. We found that hard negative mining leads to a significant increase in quality on downstream tasks in comparison to random negative sampling.

**Comparison with baselines.** We compared CoLES with baselines described in Section 4.1 in two scenarios. First, we compared embeddings produced by the CoLES encoder with other types of embeddings and with manually created aggregates by using them as input features of a downstream task model. The downstream task model is trained by LightGBM (Ke et al., 2017) independently from the sequence encoder. As Table 2 demonstrates, our method generates sequence embeddings of sequential data that achieve strong performance results in comparison to the case of manually crafted features when used on the downstream tasks. In particular, Table 2 shows that even unsupervised CoLES embeddings perform on par and sometimes even better than hand-crafted features. Also note, that CoLES embeddings outperform embeddings produced by the other self-supervised baselines on each dataset.

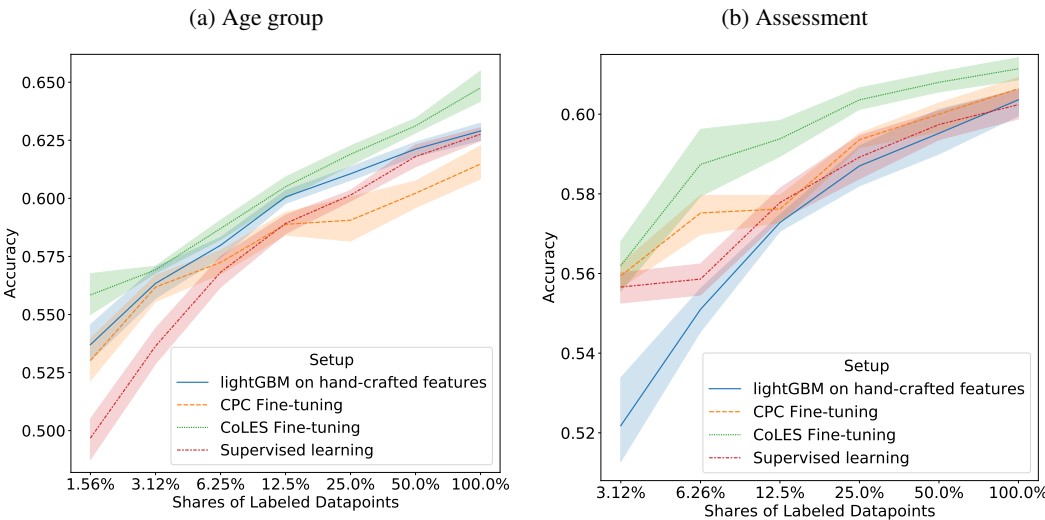

Figure 2: Model quality for different dataset sizes
The rightmost point corresponds to all labels and supervised setup.

Table 2: Accuracy on the downstream tasks: Metric increase against baseline

| Method | Age group Accuracy | Churn AUROC | Assessment Accuracy | Retail Accuracy |
|---|---|---|---|---|
| **LightGBM:** | | | | |
| **Designed features** | $0.631 \pm 0.004$ | $0.825 \pm 0.005$ | $0.602 \pm 0.006$ | $0.547 \pm 0.001$ |
| **SOP embeddings** | $-21.9\% \pm 0.6\%$ | $-5.3\% \pm 0.8\%$ | $-4.1\% \pm 1.0\%$ | $-22.8\% \pm 0.2\%$ |
| **NSP embeddings** | $-1.5\% \pm 0.9\%$ | $+0.6\% \pm 0.7\%$ | $-3.5\% \pm 1.1\%$ | $-22.3\% \pm 0.4\%$ |
| **RTD embeddings** | $+0.1\% \pm 0.6\%$ | $-2.9\% \pm 0.8\%$ | $-3.6\% \pm 1.1\%$ | $-5.0\% \pm 0.3\%$ |
| **CPC embeddings** | $-5.9\% \pm 0.6\%$ | $-2.9\% \pm 0.6\%$ | $-2.3\% \pm 0.9\%$ | $-4.0\% \pm 0.3\%$ |
| **CoLES embeddings** | $+1.1\% \pm 1.2\%$ | $\mathbf{+2.2}\% \pm 0.6\%$ | $-0.1\% \pm 0.9\%$ | $-1.4\% \pm 0.2\%$ |
| **Supervised learning** | $0.628 \pm 0.005$ | $0.817 \pm 0.012$ | $0.602 \pm 0.006$ | $0.542 \pm 0.001$ |
| **RTD fine-tuning** | $+1.2\% \pm 1.2\%$ | $+0.3\% \pm 1.3\%$ | $-2.7\% \pm 1.0\%$ | $+0.5\% \pm 0.4\%$ |
| **CPC fine-tuning** | $-2.1\% \pm 1.6\%$ | $-0.9\% \pm 1.4\%$ | $+0.7\% \pm 1.1\%$ | $+1.2\% \pm 0.3\%$ |
| **CoLES fine-tuning** | $\mathbf{+2.5}\% \pm 1.0\%$ | $\mathbf{+1.1}\% \pm 1.3\%$ | $\mathbf{+2.2}\% \pm 1.1\%$ | $\mathbf{+1.9}\% \pm 0.2\%$ |

test set quality metric $\pm 95\%$ is shown

In the second scenario, we fine-tune pre-trained models for specific downstream tasks. The models are pre-trained using CoLES and other self-supervised learning approaches and then are additionally trained on the labeled data for the specific task in the same way as we trained a neural net for the supervised learning (see Section 4.1). A neural net without pre-training is also added to the comparison. As Table 2 shows, fine-tuned representations obtained by our method achieve superior performance on all the considered datasets, outperforming all other methods by statistically significant margins.

**Semi-supervised setup.** To evaluate our method in case of the restricted amount of labeled data, we performed the series of experiments where only a fraction of available labels are used to train the downstream task model. As in the case of the supervised setup, we compare the proposed method with LigthGBM over hand-crafted features, CPC, and supervised learning without pre-training (see Section 4.1).

The results of this comparison are presented in Figure 2. Note that the difference in performance between CoLES and supervised-only methods increases as we decrease the number of available labels. Also note that CoLES consistently outperforms CPC for different volumes of labeled data.

**Business applications.** In addition to the described experiments on public datasets, we have performed extensive testing of our method on the private data in a large European bank. We've observed a significant increase in model performance (+ 2-10% AUROC) after the addition of CoLES embeddings to the existing models in many downstream tasks, including credit scoring, marketing campaign targeting, product recommendations cold start, fraud detection and legal entities connections prediction.

## 5 CONCLUSIONS

In this paper, we present *Contrastive Learning for Event Sequences (CoLES)*, a novel self-supervised method for building embeddings of discrete event sequences. In particular, the CoLES method can be effectively used for pre-training neural networks in semi-supervised settings. It can also be used to produce embeddings of complex event sequences that can be effectively used in various downstream tasks.

We also empirically demonstrate that our approach achieves strong performance results on several downstream tasks and consistently outperforms both classical machine learning baselines on handcrafted features, as well as other previously introduced self-supervised and semi-supervised learning baselines adapted to the event sequence domain. In the semi-supervised setting, where the number of labeled data is limited, our method demonstrates even stronger results: the lesser is the labeled data the larger is performance margin between CoLES and supervised-only methods.

The method is especially adapted for event sequence data which is extensively used by the core businesses of many large companies, including financial institutions, internet companies, retail and telecom.

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

## A    BATCH GENERATION

We consider several empirical strategies for the sub-sequence generation to compare with the random slices algorithm described in section "Sampling of surrogate sequences" of the main paper. The simplest strategy is random sampling without replacement. One more strategy is to produce sub-sequences by the random splitting of the initial sequence to several connected segments without an intersection between them. To generate k sub-sequences, the following procedure should be repeated k times: take a random number of elements from the sequence *without replacement*.

---
**Algorithm 2:** Disjointed sub-sequences generation strategy

---
**hyperparameters:** $k$: number of sub-sequences to be produced.
**input:** A sequence $S$ of length $l$.
**output:** $S_1, ..., S_k$: sub-sequences of $S$.

Generate vector $inds$ of length $l$ with random integers from [1,k].
**for** $i \leftarrow 1$ **to** $k$ **do**
    | $S_i = S[inds == i]$
**end**

---

## B    PERFORMANCE OPTIMIZATIONS

Here we describe several performance optimizations that can be used for model training and inference.

**Pair distance calculation.** In order to select negative samples, we need to compute pair-wise distance between all possible pairs of embedding vectors of a batch. For the purpose of making this procedure more computationally effective we perform normalization of the embedding vectors, i.e. project them on a hyper-sphere of unit radius. Since $D(A, B) = \sqrt{\sum_i (A_i - B_i)^2} = \sqrt{\sum_i A_i^2 + \sum_i B_i^2 - 2\sum_i A_i B_i}$ and $||A|| = ||B|| = 1$, to compute the euclidean distance we only need to compute: $\sqrt{2 - 2(A \cdot B)}$.

To compute the dot product between all pairs in a batch we just need to multiply the matrix of all embedding vectors of a batch by itself transposed, which is a highly optimized computational procedure in most modern deep learning frameworks. Hence, the computational complexity of the negative pair selection is $O(n^2 h)$ where $h$ is the size of the output embeddings and $n$ is the size of the batch.

**Embedding update calculation.** Encoder, based on RNN-type architecture like GRU (Cho et al., 2014), allows to calculate embedding $c_{t+k}$ by updating embedding $c_t$ instead of calculating embedding $c_{t+k}$ from the whole sequence of past events $z_{1:t}$: $c_k = rnn(c_t, z_{t+1:k})$. We use this

optimization to reduce inference time to update already existing person embeddings with new events, occurred after the calculation of embeddings. This is possible due to the recurrent nature of RNN-like networks.

## C    PROOF OF THEOREM 1

In this section, we provide proof of Theorem 1 (Section 3.3) that justifies the Random Slices sub-sequence generation strategy proposed in the paper.

*Proof.* First, we state the following straightforward lemma:

**Lemma 1.** *Let a stochastic process $\{Y(t)\}_{t=1}^{\infty}$ be a shift of another stochastic process $\{\widehat{Y}(t)\}_{t=1}^{\infty}$ by independent random time $s$, i.e. $Y(t) = \widehat{Y}(t + s)$ with integer $s \geq 0$. If process $\{\widehat{Y}(t)\}_{t=1}^{\infty}$ is cyclostationary with period $\widehat{T}$ and $(s_e \mod \widehat{T})$ is uniform over $[0, \widehat{T} - 1]$, then process $\{Y(t)\}_{t=1}^{\infty}$ is stationary.*

Lemma 1 implies that process $\{X_e(t)\}_{t=1}^{T_e}$ is stationary, and all its segments $\{X_e(t)\}_{t=s'+1}^{T'_e+s'}$ of a given length $T'_e$ define the same distribution over sequences as its starting segment $\{X_e(t)\}_{t=1}^{T'_e}$ does. Furthermore, integrating over $s'$, we conclude that the conditional distribution of a sub-sequence obtained via Random Slices generation strategy given its length $T'_e$ follows the process $\{X_e(t)\}_{t=1}^{T'_e}$. To finish the proof, it remains to prove Equation 1.

Assume $\mathbb{P}(T_e = k) \propto k^{\alpha}$ for $k \in [m, \infty]$. By the law of total probability, we have $\mathbb{P}(T'_e = k_0) = \sum_k \mathbb{P}(T_e = k)\mathbb{P}(T'_e = k_0 \mid T_e = k)$, that is,

$$\mathbb{P}(T'_e = k_0) = C \sum_{k=k_0}^{\infty} k^{\alpha-1},$$

where $C$ is the normalization constant. To estimate the sum of the series, notice that

$$\int_{k_0-\frac{1}{2}}^{\infty} x^{\alpha-1}dx > \sum_{k=k_0}^{\infty} k^{\alpha-1} > \int_{k_0}^{\infty} x^{\alpha-1}dx, \tag{2}$$

where the former inequality follows from the fact that $\int_{k-1/2}^{k+1/2} x^{\alpha-1} > k^{\alpha-1}$ as long as function $f(x) = x^{\alpha-1}$ is convex. After integration, we rewrite Equation 2 as follows:

$$\frac{-1}{\alpha}\left(k_0 - \frac{1}{2}\right)^{\alpha} > \sum_{k=k_0}^{\infty} k^{\alpha-1} > \frac{-1}{\alpha}k_0^{\alpha}. \tag{3}$$

Using these inequalities, we obtain the upper bound for $\mathbb{P}(T'_e = k)$ in the following way:

$$\mathbb{P}(T'_e = k_0) = \sum_{k=k_0}^{\infty} k^{\alpha-1} / \sum_{l=m}^{\infty}\sum_{k=l}^{\infty} k^{\alpha-1} < \frac{-1}{\alpha}\left(k_0 - \frac{1}{2}\right)^{\alpha} / \sum_{l=m}^{\infty} \frac{-1}{\alpha}l^{\alpha} =$$

$$= \left(\frac{k_0}{k_0 - 1/2}\right)^{-\alpha} k_0^{\alpha} / \sum_{l=m}^{\infty} l^{\alpha} = \left(\frac{k_0}{k_0 - 1/2}\right)^{-\alpha} \mathbb{P}(T_e = k_0).$$

At last, the lower bound for $\mathbb{P}(T'_e = k)$ can be obtained using Equation 3 as follows:

$$\mathbb{P}(T'_e = k_0) = \sum_{k=k_0}^{\infty} k^{\alpha-1} / \sum_{l=m}^{\infty}\sum_{k=l}^{\infty} k^{\alpha-1} > k_0^{\alpha} / \sum_{l=m}^{\infty}\left(l - \frac{1}{2}\right)^{\alpha} >$$

$$> \left(\frac{m - 1/2}{m}\right)^{-\alpha} k_0^{\alpha} / \sum_{l=m}^{\infty} l^{\alpha} = \left(\frac{m - 1/2}{m}\right)^{-\alpha} \mathbb{P}(T_e = k_0).$$

The latter inequality in these calculations follows from the fact that $\left(\frac{l-\frac{1}{2}}{l}\right)^{\alpha} < \left(\frac{m-\frac{1}{2}}{m}\right)^{\alpha}$ for $l > m$.

$\square$

Table 3: Data structure for a single credit card

| Date | Time | Amount | Currency | Country | Merchant Type |
|---|---|---|---|---|---|
| Jun 21 | 16:40 | 230 | EUR | France | Restaurant |
| Jun 21 | 20:15 | 5 | USD | US | Transportation |
| Jun 22 | 09:30 | 40 | USD | US | Household Appliance |

Table 4: Click-stream structure for a single user

| Time | Date | Domain | Referrer Domain |
|---|---|---|---|
| 17:40 | Jun 21 | amazon.com | google.com |
| 17:41 | Jun 21 | amazon.com | amazon.com |
| 17:45 | Jun 21 | en.wikipedia.org | google.com |

## D  DATASETS

We designed the method specially for the user behavior sequences (Ni et al., 2018). These sequences consist of discrete events per person in continuous time, for example, behavior on websites, credit card transactions, etc.

Considering credit card transactions, each transaction has a set of attributes, either categorical or numerical including the timestamp of the transaction. An example of the sequence of three transactions with their attributes is presented in Table 3. The merchant type field represents the category of a merchant, such as "airline", "hotel", "restaurant", etc.

Another example of user behavior data is click-stream: the log of internet page visits. The example of a click-stream log of a single user is presented in Table 4.

In our research we chose several publicly available datasets from data science competitions.

1. **Age group prediction competition**[7] - the task is to predict the age group of a person. The group ratio is balanced in the dataset. The dataset consists of 44M anonymized transactions representing 50k persons with a target labeled for only 30k of them (27M out of 44M transactions), for the other 20k persons (17M out of 44M transactions) label is unknown. Each transaction includes date, type (for example, grocery store, clothes, gas station, children's goods, etc.) and amount. We use all available 44M transactions for contrastive learning, excluding 10% - for the test part of the dataset, and 5% for the contrastive learning validation.

2. **Churn prediction competition**[8]. The dataset of 1M anonymized card transactions representing 10K clients was used to predict a churn probability. 5k clients have labels (0.49M out of 1M transactions), 5.2k clients haven't labels (0.52M out of 1M transactions). Target is binary, almost balanced with proportions 0.55 and 0.45. Transactions of the same type and month are grouped and represented as a single pseudo-transaction, which amount is the sum of grouped transactions. Each transaction is characterized by date, type, amount and Merchant Category Code.

3. **Assessment prediction competition**[9] - the task is to predict the results of the in-game assessment based on the history of children gameplay data. Target is one of 4 grades, with proportions 0.50, 0.24, 0.14, 0.12. The dataset consists of 12M gameplay events combined in 330k gameplays representing 18k children. 17.7k gameplays (0.9M out of 12M gameplay events) are labeled, the remaining 312k gameplays (11.6M out of 12M gameplay events) are not labeled. Each gameplay event is characterized by timestamp, event code, the incremental counter of events within a game session, time since the start of the game session, etc.

---

[7]https://onti.ai-academy.ru/competition

[8]https://boosters.pro/championship/rosbank1/

[9]https://www.kaggle.com/c/data-science-bowl-2019

Figure 3: Event sequence length distribution

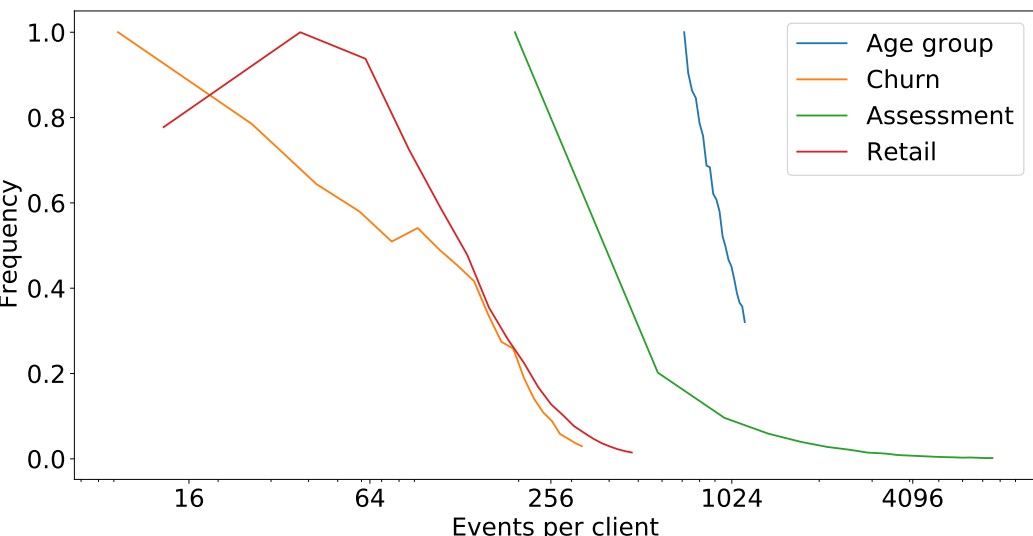

4. **Retail purchase history age group prediction**[10] - the task is to predict the age group of a client based on its retail purchase history. The group ratio is balanced in the dataset. The dataset consists of 45,8M retail purchases representing 400k clients. Only labeled data is used. Each purchase is characterized by time, product level, segment, amount, value, points received.

To check that sequence lengths of the considered datasets follow the power-low distribution we measured their distribution of lengths. As Figure 3 shows, the event sequence length distribution is close to the power-law distribution for every considered dataset.

To check that considered datasets follow our repeatability and periodicity assumption made in Section 3.2 and used for theoretical analysis in Section 3.3 we performed the following experiments. We measure the KL-divergence two kinds of samples: (1) between random sub-samples of the same sequence, generated using a modified version of Algorithm 1 where overlapping events are dropped and (2) between random sub-samples taken from different sequences. The results are shown in Figure 4. As Figure 4 shows, the KL-divergence between sub-sequences of the same sequence of events is relatively small compared to the typical KL-divergence between sub-samples of different sequences of events. This observation supports our repeatability and periodicity assumption. Also note that additional plot (e) is provided as an example for data without any repeatable structure.

## E EXPERIMENT SETUP

For all methods, a random search on 5-fold cross-validation over the train set is used for hyper-parameter selection. The hyper-parameters with the best out-of-fold performance on the train set are then chosen. The final set of hyper-parameters used for CoLES is shown in Table 5. The number of sub-sequences generated for each sequence was always 5 for each dataset.

### E.1 HAND-CRAFTED FEATURES

Here we describe the details of producing hand-crafted features. All attributes of each transaction are either numerical (e. g. amount) or categorical (e.g. merchant type (MCC code), transaction type, etc.). For the numerical type of attribute we apply aggregation functions, such as 'sum', 'mean', 'std', 'min', 'max', over all transactions per user. For example, if we apply 'sum' for the numerical field 'amount'

---

[10]https://ods.ai/competitions/x5-retailhero-uplift-modeling

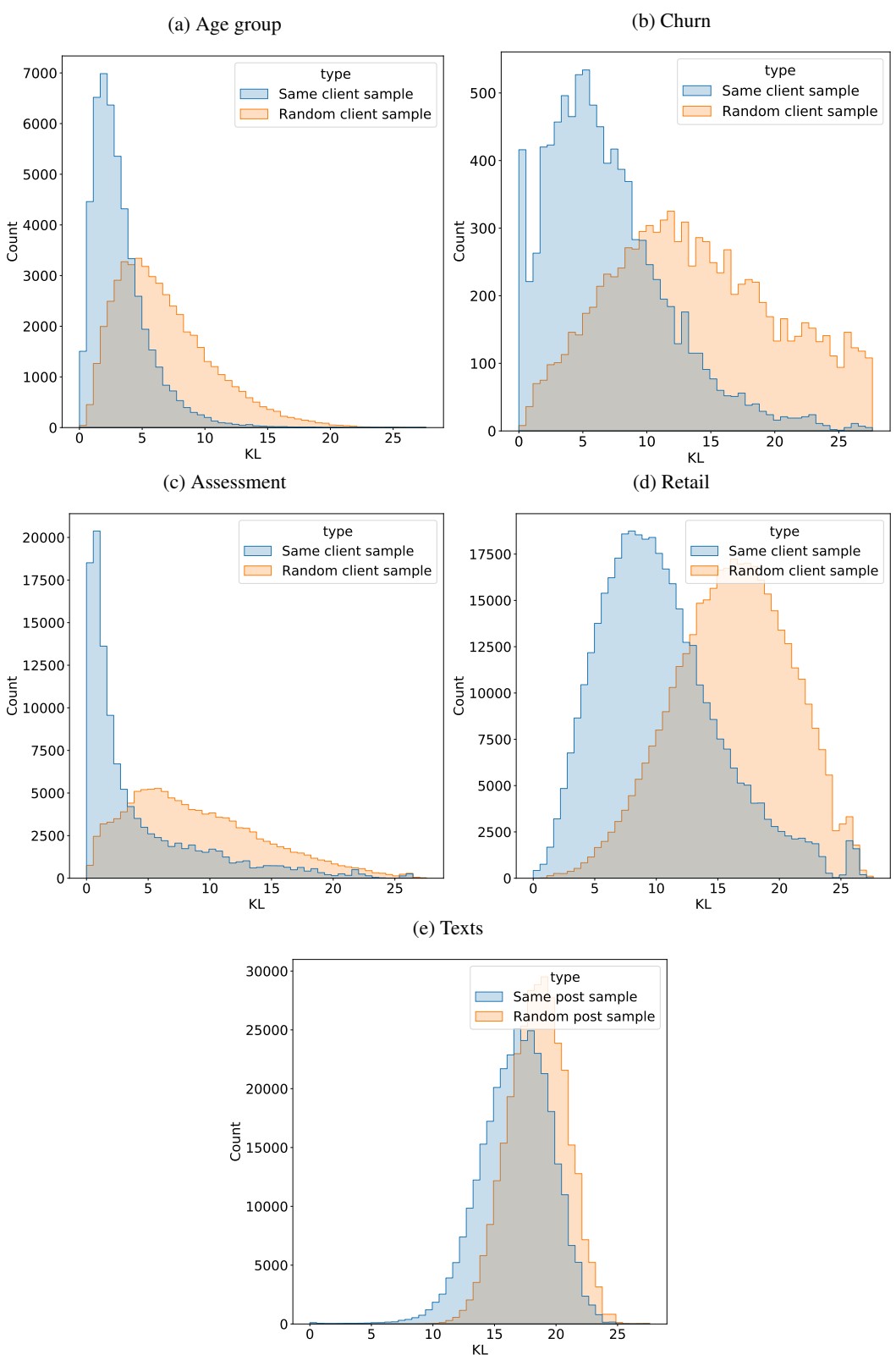

Figure 4: Periodicity and repeatbility of the data. KL-divergence between event types of two random sub-sequences from the same sequence is compared with KL-divergence between sub-sequences of different sequences.

Table 5: Hyper-parameters for CoLES training

| Dataset | Output size | Learning rate | N samples in batch | N epochs | Min seq length | Max seq length | Encoder |
|---------|-------------|---------------|--------------------|----------|----------------|----------------|---------|
| **Age group** | 800 | 0.001 | 64 | 150 | 25 | 200 | GRU |
| **Churn** | 1024 | 0.004 | 128 | 60 | 15 | 150 | LSTM |
| **Assessment** | 100 | 0.002 | 256 | 100 | 100 | 500 | GRU |
| **Retail** | 800 | 0.002 | 256 | 30 | 30 | 180 | GRU |

Table 6: Comparison of encoder types

| Dataset | LSTM | GRU | Transformer |
|---------|------|-----|-------------|
| **Age group** (Accuracy) | $0.621 \pm 0.008$ | **0.638** $\pm 0.007$ | $0.622 \pm 0.006$ |
| **Churn** (AUROC) | **0.823** $\pm 0.017$ | $0.812 \pm 0.010$ | $0.780 \pm 0.012$ |
| **Assessment** (Accuracy) | **0.620** $\pm 0.007$ | $0.618 \pm 0.009$ | $0.542 \pm 0.007$ |
| **Retail** (Accuracy) | $0.535 \pm 0.003$ | **0.542** $\pm 0.002$ | $0.499 \pm 0.002$ |

5-fold cross-validation metric $\pm 95\%$ is shown

we obtain a feature 'sum of all transaction amounts per user'. For the categorical type of attribute we apply aggregation functions in a slightly different way. For each unique value of categorical attribute we apply aggregation functions, such as 'count', 'mean', 'std' over all transactions per user' numerical attribute. For example, if we apply 'mean' for the numerical attribute 'amount' grouped by categorical attribute 'MCC code' we obtain a feature 'mean amount of all transactions for each MCC code per user'. For example, for age prediction task we have one categorical attribute (small group) with 200 unique values, combining it with amount we can produce $200 * 3$ features ('group0 x amount x count', 'group1 x amount x count', ..., 'group199 x amount x count', 'group0 x amount x mean', ...). In total we use approx 605 features for this task. Note, that hand-crafted features contain information about user spending profile but omit information about transactions temporal order.

## F    RESULTS

### F.1    DESIGN CHOICES OBSERVATIONS

We consider several contrastive learning losses that showed promising performance on different datasets (Kaya and Şakir Bilge, 2019) and some classical variants: contrastive loss (Hadsell et al., 2006), binomial deviance loss (Yi et al., 2014), triplet loss (Hoffer and Ailon, 2015), histogram loss (Ustinova and Lempitsky, 2016), and margin loss (Manmatha et al., 2017). The results of comparison are shown in the Table 7.

It is interesting to observe that even contrastive loss that can be considered as the basic variant of contrastive learning loss allows to get strong results on the downstream tasks (see Table 7). Our hypothesis is that an increase in the model performance on contrastive learning task does not always lead to an increase in performance on downstream tasks.

As shown in Table 6, different choices of encoder architectures show comparable performance on the downstream tasks.

### F.2    EMBEDDING SIZE

Figure 5 shows that the performance quality on the downstream task increases with the dimensionality of an embedding. After the best quality is achieved, a further increase in the dimensionality of an embedding dramatically reduces quality. These results can be interpreted as the bias-variance trade-off. When the embedding dimensionality is too small, too much information can be discarded (high bias). On the other hand, when embedding dimensionality is too large, too much noise is added (high variance). Note, that increasing the embedding size will also linearly increase the training time and the volume of consumed memory on the GPU.

Figure 5: Embedding dimensionality vs. quality

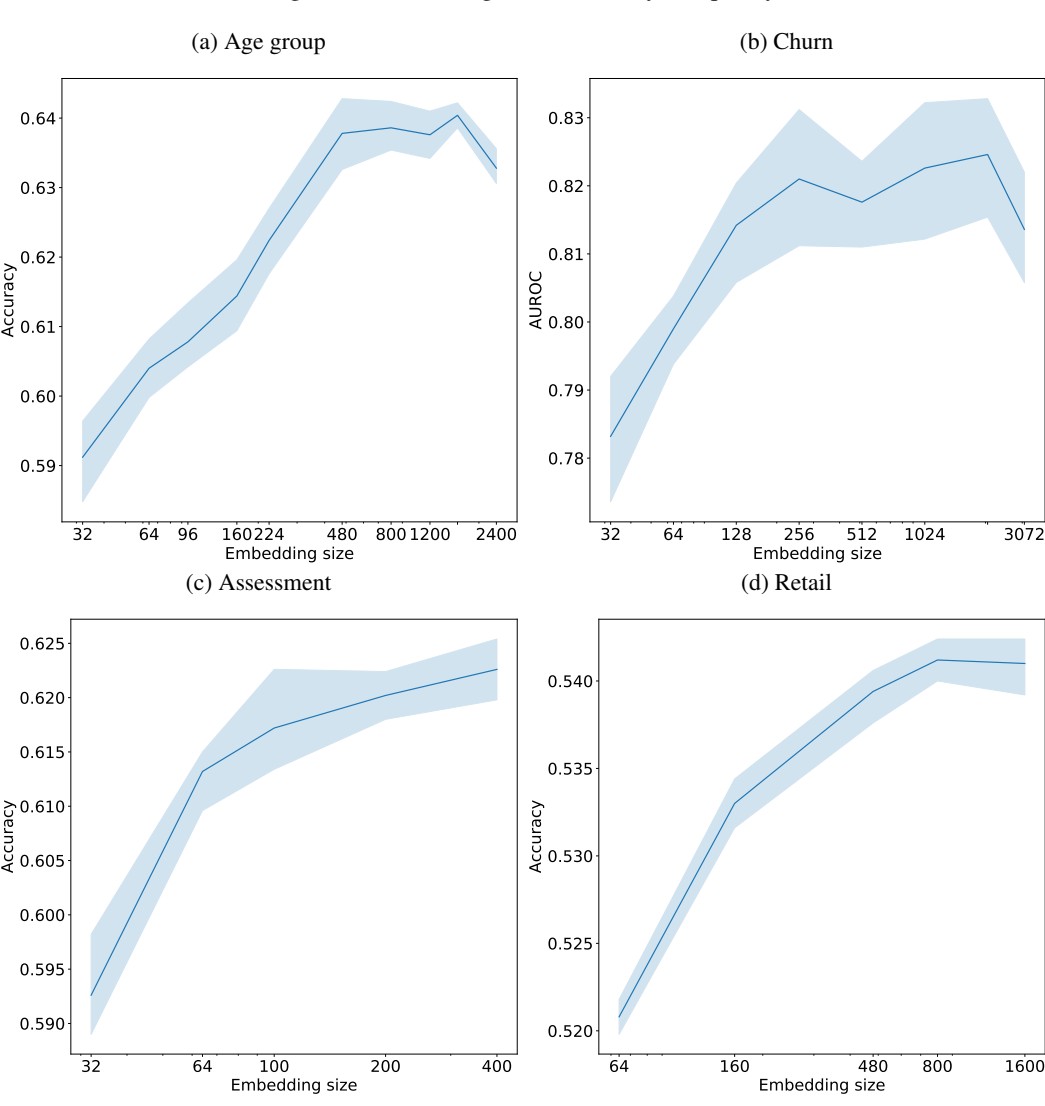

Table 7: Comparison of contrastive learning losses

| Dataset | Contrastive (margin=0.5) | Binomial deviance | Histogram | Margin | Triplet |
|---|---|---|---|---|---|
| **Age group** | **0.639** | 0.621 | 0.632 | 0.638 | 0.636 |
| (Accuracy) | ±0.006 | ±0.005 | ±0.008 | ±0.007 | ±0.004 |
| **Churn** | **0.823** | 0.769 | 0.815 | **0.823** | 0.781 |
| (AUROC) | ±0.017 | ±0.018 | ±0.018 | ±0.012 | ±0.021 |
| **Assessment** | **0.618** | 0.589 | 0.615 | 0.612 | 0.600 |
| (Accuracy) | ±0.009 | ±0.004 | ±0.007 | ±0.005 | ±0.004 |
| **Retail** | **0.542** | 0.535 | 0.533 | 0.541 | 0.541 |
| (Accuracy) | ±0.002 | ±0.004 | ±0.002 | ±0.001 | ±0.001 |

5-fold cross-validation metric $\pm 95\%$ is shown

Table 8: Comparison of negative sampling strategies

| Dataset | Hard negative mining | Random negative sampling | Distance weighted sampling |
|---|---|---|---|
| **Age group** (Accuracy) | **0.639** ±0.006 | 0.626 ± 0.008 | 0.629 ± 0.004 |
| **Churn** (AUROC) | **0.823** ±0.017 | 0.815 ± 0.013 | 0.821 ± 0.014 |
| **Assessment** (Accuracy) | **0.618** ±0.009 | 0.593 ± 0.002 | 0.603 ± 0.010 |
| **Retail** (Accuracy) | **0.542** ±0.002 | 0.530 ± 0.002 | 0.536 ± 0.002 |

5-fold cross-validation metric $\pm 95\%$ is shown

## F.3 SEMI-SUPERVISED SETUP

To evaluate our method in case of the restricted amount of labeled data, we use only part of the available target labels for the semi-supervised experiment, see Section 4.2 for details. As in the case of the supervised setup, we compare the proposed method with LigthGBM over hand-crafted features, CPC, and supervised learning without pre-training. In figure 6 we provide learning curves for all considered datasets.

## F.4 EMBEDDING VISUALIZATION

In order to visualize CoLES embeddings in 2-dimensional space, we applied tSNE transformation (van der Maaten and Hinton, 2008) on them. tSNE transforms high-dimensional space to low-dimensional based on local relationships between points, so neighbor vectors in high-dimensional embedding space are pushed to be close in 2-dimensional space. We colorized 2-dimensional vectors using the target values of the datasets.

Note, that embeddings was learned in a fully self-supervised way from raw user transactions without any target information. The sequence of transactions represent user' behavior, thus the CoLES model captures behavioral patterns and outputs embeddings of users with similar patterns nearby. As shown below, local clusters in embedding space correspond to the distribution of user's attributes either age or churn fact.

tSNE vectors from the age prediction dataset are presented in Figure 7a. We can observe 4 clusters: clusters for group '1' and '2' are on the opposite side of the cloud, clusters for groups '2' and '3' are in the middle.

Taking into account that age is an ordinal attribute, we can make an assumption about the ordering of age groups: $age(1) < age(3) < age(0) < age(2)$ or vice versa. ($age(bin)$ returns age of user for specific group).

tSNE points from the churn prediction dataset are presented in Figure 7b. There are areas where one type of label dominates over the other.

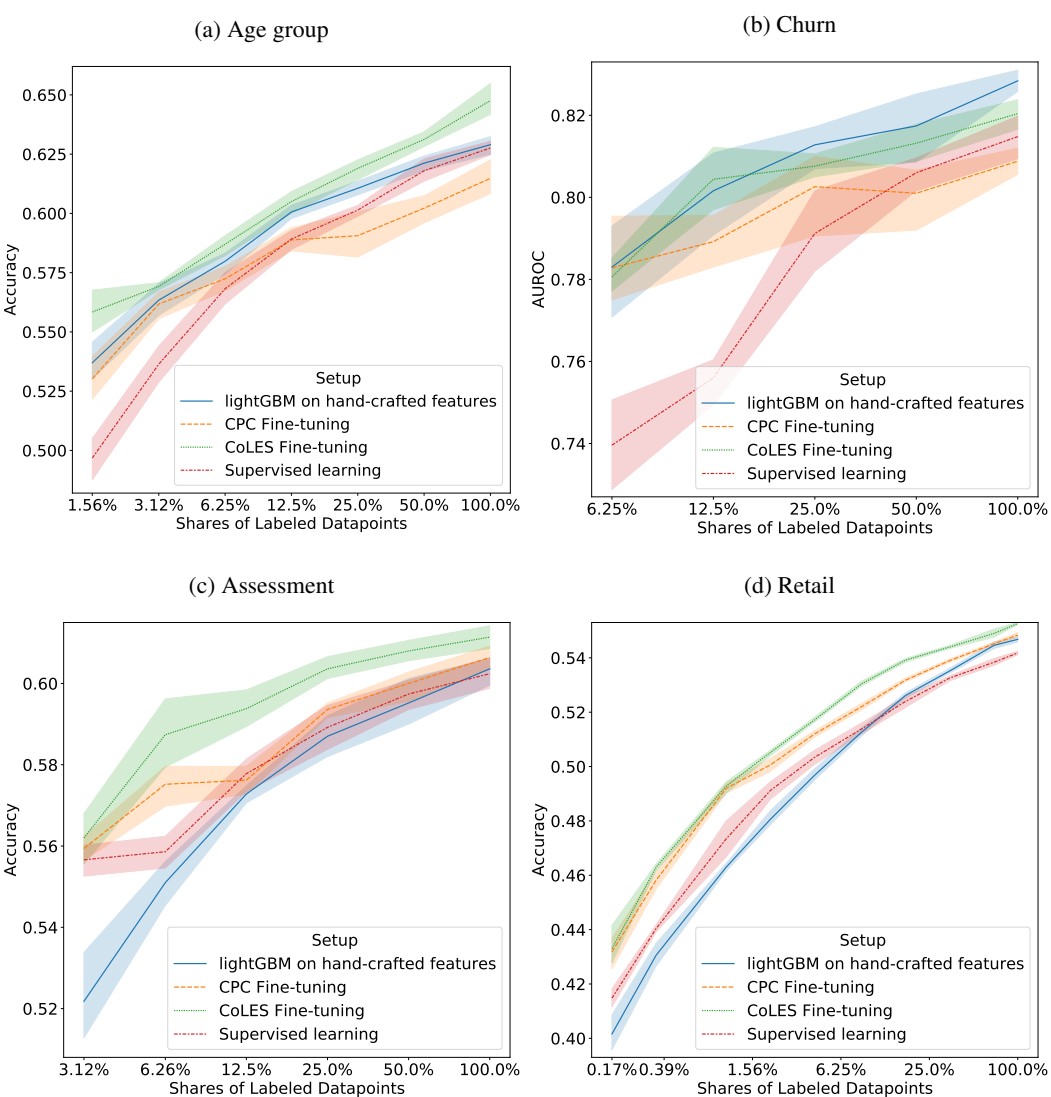

Figure 6: Model quality for different dataset sizes
The rightmost point corresponds to all labels and supervised setup.

Figure 7: 2D tSNE mapping of CoLES embeddings colored by target labels

(a) Age group

(b) Churn

(c) Assessment

(d) Retail

