# OpenReview forum: "CoLES: Contrastive learning for event sequences with self-supervision"
_ICLR.cc/2021/Conference — Reject_

### Official Review · AnonReviewer4 · 2020-10-27
**Simple and powerful augmentation strategy for sequential data**

**Rating:** 5
**Confidence:** 3

**Review:**

Self-supervised learning is a rapidly growing domain where the amount of labeled data is limited. Contrastive learning with data augmentation enables us to design a loss function which makes semantically similar objects closer to each other, while dissimilar ones further away. The authors argue that self-supervised learning approaches have been focused on NLP or Computer vision, rather than sequential user behaviors datasets (mostly tabular). In order to address this problem, this paper proposed a random slices subsequence generation strategy (CoLES) for contrastive learning in event sequences.

The proposed CoLES algorithm is a data augmentation strategy that randomly selects subsequences from the full event sequence from each user. By theoretical analysis, the authors claimed that subsequences generated by CoLES are representative samples of the original entity and follows its latent distribution. I am not sure about it (See the end of this review).

The CoLES algorithm is tested with several publicly available datasets: 1) Age group prediction, 2) Churn prediction, 3) Assessment prediction, 4) Retail purchase history age group prediction, these datasets include sufficient amounts of discrete events per user. It was not clearly stated how CoLES was used to solve the classification (1, 3, 4) and regression (2) problems. However, considering the contents of chapter 3.4-3.5, I was able to infer that K subsequences are created with CoLES as well as several negative samples. Then each subsequence is encoded by the event encoder, then, the contrastive loss is minimized together with the final classification or regression using the sequence encoder (RNN) described in chapter 3.5. Describing the overall procedure will help readers to understand the framework better.

Through various experiments, the authors show that the CoLES sampling strategy is superior. (Table 1) Compared with baseline sampling strategies for sequential data, (Figure 2) Augmentation by COLES is helpful in small-labeled SSL scenarios, (Table 2)  Pretrained model using COLES performs is effective on fine-tuning stage for downstream classification. Supplementary results such as changing encoder types, contrastive learning losses, negative sampling strategies, embedding sizes, full-results on dataset sizes are also available in the appendix.

The CoLES algorithm is very simple and intuitive. Although there is no technical advance,  CoLES showed a strong performance and it seems that CoLES can be widely used because of its simplicity. But, I still have some concerns about the experiments that the performance was compared only within the proposed subsequence augmentation problem setting, so it was not possible to find out how excellent CoLES is in the global solution space. For example, training a model by using additional augmented data generated by CTGAN [1], or tabular learning framework by masking some entities [2] can be strong baselines for wider experiment scope.

- [1] CTGAN: https://github.com/sdv-dev/CTGAN
- [2] TabNet: https://github.com/dreamquark-ai/tabnet


The contents are easy to follow, but the overall presentation should be improved. Here are Some typos and issues that I found:
- Footnote 1: LeCunn -> LeCun
- Page 7: self-superivsed -> self-supervised
- Page 8: pre-traning -> pre-training
- Page 8: Descriptions/Results of real-world business applications can be removed / or can be explained in a little more detail in the Appendix.
- Page 9-11: Reference format does not look very natural
- Page 12-13: The notation for probability is broken.
- Page 14: Numbering of bullet points, ???M retail purchases representing ???k clients
- Page 15: Event sequence lenght  -> Event sequence length (Also, it is difficult to say that event sequence length follows power-law distribution)
- Figure 4 on Page 17: Would be better to have equally-spaced x-ticks.
- Page 19: Terse writing

About the theorem: To prove that the subsequences are representative samples, the upper- and lower-bounds of the conditional probability of the length of the subsequences were calculated, with underlying three assumptions. But, the reviewer could not understand why this theorem proved the above claim. So, I want to leave the verification of this proof to other reviewers.

---

> ### Author Response · Authors · 2020-11-18
> **Answers to the third reviewer**
>
> We are very grateful to you for your attentive reading and important comments.  We are also really grateful for mentioning different typos in our paper. We address your questions below:
>
> > It was not clearly stated how CoLES was used to solve the classification (1, 3, 4) and regression (2) problems.
>
> We have described it in section 4.2 with the experiment's result in “Comparison with baselines” paragraph. But we will add it to section 3 (overview of the CoLES method), to make it easier to understand for the readers.
>
> In our work, we use two different scenarios for applying our method to the downstream problems. In the first scenario, we use the representations (obtained from the encoder of CoLES method) as features for training LightGBM. In this scenario, we compare our method with LightGBM based on handcrafted features and on representations obtained from other self-supervised methods.
>
> In the second scenario, we take the encoder of CoLES and fine-tune it for the downstream tasks. In this case, we compare our method with other fine-tuned models, pre-trained with other self-supervised methods. We also compare it with a supervised model trained without pre-training.
>
> Note that all of the downstream tasks considered in the paper are classification tasks, including (2), but our method can be applied to a regression problem as well.
>
> > But, I still have some concerns about the experiments that the performance was compared only within the proposed subsequence augmentation problem setting, so it was not possible to find out how excellent CoLES is in the global solution space. For example, training a model by using additional augmented data generated by CTGAN [1], or tabular learning framework by masking some entities [2] can be strong baselines for wider experiment scope.
>
> Thank you for mentioning other algorithms that we can consider in our study. However, we do not think these methods are directly applicable to our setting of event sequences. Despite the fact that each event can be considered as a row in a table, one sample in the dataset is not an event, but a sequence of events. The aim of the considered downstream tasks is to classify the whole sequence, hence we have to consider methods that can be applied to sequences. Please note that both CTGAN and TabNET are designed for tabular data, and hence, can only be applied to separate events but not to sequences of events.
>
> Also, note that the main reason for using data augmentations was to generate pairs of similar sequences for the self-supervised contrastive learning task and they are of limited value for supervised training scenarios.
>
> The data augmentation strategy is only a part of the proposed algorithm, which we called CoLES. We use this augmentation strategy to sample positive pairs of subsequences from a sequence. We sample negative pairs from different sequences. Next, we use obtained pairs for contrastive learning to learn sequence representations in a self-supervised way. Those representations can then be used as features for self-supervised downstream tasks or, alternatively, one can fine-tune the encoder network weights trained to generate self-supervised representations.
>
> > But, the reviewer could not understand why this theorem proved the above claim. So, I want to leave the verification of this proof to other reviewers.
>
> We apologize that some notations were broken in the proof of Theorem 1. We reduced Theorem 1 to calculations of the conditional probability of the length using Lemma 1 and the paragraph right below its formulation. This part is rather straightforward, and the most interesting part of Theorem 1 is that the distribution of length remains almost that same despite that a sampled subsequence is shorter than the initial sequence.

---

### Official Review · AnonReviewer1 · 2020-10-27
**This paper uses a self-supervised learning based approach to learn efficient representations for downstream discrete event sequence prediction domains.**

**Rating:** 5
**Confidence:** 4

**Review:**

Pros:

1. The paper targets the problem of event sequence prediction in a contrastive self-supervised learning framework .They train this contrastive learning method by generating positive and negative samples via data augmentation method proposed as random slicing, which creates overlapping sub-sequences from each task sequence. This method follows the same distribution and properties as the original sequence which is supported by proofs.
2.This approach shows improvements in the semi-supervised setting and supervised setting compared to other self-supervised training baselines and also against hand-crafted feature generation techniques.
3.The problem is formulated properly that it is easy to follow.

Cons:

1.The paper applies the ideas of data augmentation, random sampling and self-supervised learning in the event sequence prediction domain. Although these strategies may have been used first time in this setting, they have minor contributions in these individual methods.
2.The framework shown in Fig 1., is a very classic contrastive learning based approach which has gained popularity in self-supervised literature as well. When the authors use CPC as a baseline, do they consider the same data augmentation and random sampling techniques as shown in Fig1.?  It is not very convincing that CPC will perform worse than the regular contrastive loss function.
3.Are the final embeddings fine-tuned when they are trained for the downstream task or are they fixed ?
4.As accuracy is being used as a metric in the paper, it is not very clear what are the number of classes for all datasets used. It will be good to add this.
5.Why is the accuracy negative in Table2, is it the relative improvement with respect to the supervised setting? The authors should put absolute accuracy instead of relative improvement for each.
6.Is the supervised model also trained with data augmentation? Self-supervised doing better than supervised is not so intuitive and could you discuss why the supervised performance in general is low for these tasks ?
7.The paper has a lot of information as a laundry list that makes it harder to find the motivation and intuition behind different design choices. I will highly recommend to restructure the paper with stronger motivation on the choice of model in fig 1.

####### Post rebuttal #########

I thank the authors for their well formed responses. They address all my concerns effectively. I keep my original rating and recommendations.

---

> ### Author Response · Authors · 2020-11-18
> **Supervised vs self-supervised setup**
>
> > Self-supervised doing better than supervised is not so intuitive and could you discuss why the supervised performance in general is low for these tasks ?
>
> First of all, in all scenarios a significant part of datasets is not labeled and therefore cannot be used for training in supervised settings, therefore self-supervised methods have an advantage.
>
> Also note that we considered two different scenarios: self-supervised embeddings and fine-tuned encoder presented in the two halves of Table 2. Please note that in the first scenario we used the LightGBM model for the application of frozen self-supervised embeddings to the downstream tasks. In the second scenario, we used an encoder with one additional linear head layer for the final prediction. LightGBM is typically more powerful than a single linear layer. We believe that this can be one of the reasons why self-supervised embeddings fed to LightGBM model as features perform better than the simple neural network (see the second scenario) trained in the supervised setting.
>
> To avoid possible misunderstanding here is the brief summary of our experiment setup (also described in Section 4.1 of the paper):
>
> For the self-supervised embeddings scenario (first half of Table 2), we compared embeddings with hand-crafted features. Embeddings (or hand-crafted features, which can be considered as specific types of embeddings) were used as input variables for the supervised downstream task.
>
> For the fine-tuned encoder scenario (second half of Table 2), we used the simple neural network consisting of the encoder and an additional linear head layer. In the basic case (called supervised learning in Table 2) we had not used any pre-training for the encoder and just trained the network using supervised cross-entropy loss. In other cases, we applied self-supervised pre-training and then fine-tuned the pre-trained network using labels for the downstream task.

---

> ### Author Response · Authors · 2020-11-18
> **Answers to the second reviewer**
>
> We are very grateful to you for your attentive reading and important comments. We address your questions below:
>
> > When the authors use CPC as a baseline, do they consider the same data augmentation and random sampling techniques as shown in Fig1.? It is not very convincing that CPC will perform worse than the regular contrastive loss function
>
> This is the most important question. CPC is a learning framework based on predictive coding on the level of events, while the framework in Fig. 1 deals with (sub)sequences, and thus it cannot be directly applied to CPC. During our experiments with CPC framework, we tried certain types of augmentations in a different way, like dropping random events from the sequence or trimming the sequence. We did not observe statistically significant improvements.
>
> Moreover, we have the following explanation of why CoLES outperforms CPC. Contrastive Predictive Coding assumes that neighboring terms are highly correlated, which is satisfied in audio and computer vision domains. The goal of CPC training is to preserve mutual information between them. In an event sequence data, the mutual information between neighboring events, although positive, can be rather small, hence providing a weak signal for learning CPC.
>
>
> > The paper applies the ideas of data augmentation, random sampling and self-supervised learning in the event sequence prediction domain. Although these strategies may have been used first time in this setting, they have minor contributions in these individual methods.
>
> Most parts of the framework are indeed proposed in various other works. But this particular combination of parts was not used before. Moreover, it is not at all obvious a priori whether self-supervised contrastive learning is applicable to such important industrial domains as event sequences (see our answers to Reviewer 1). In fact, straightforward SOTA like CPC does not work well in our paper.
>
>
> > Are the final embeddings fine-tuned when they are trained for the downstream task or are they fixed ?
>
> We consider both options. The upper part of Table 2 shows the results when embeddings are frozen. The lower part of Table 2 shows results when the pre-trained encoder was fine-tuned.
>
> > As accuracy is being used as a metric in the paper, it is not very clear what are the number of classes for all datasets used. It will be good to add this.
>
> We will add the number of samples per class to the dataset descriptions at the beginning of the section “Experiments” in the revised version of the paper. Note that class proportions are almost balanced for all datasets except the “Assessment” dataset.
>
> > Why is the accuracy negative in Table2, is it the relative improvement with respect to the supervised setting? The authors should put absolute accuracy instead of relative improvement for each.
>
> As for relative numbers in the final table, we assumed that it is more convenient to see the improvement over the most popular baseline. Note that we considered two different scenarios: self-supervised embeddings and fine-tuned encoder, hence there are two methods with absolute values of accuracy, and for other methods, we show relative improvement.
>
> > Is the supervised model also trained with data augmentation?
>
> For supervised models, the same augmentations can be seen as dropping random transactions from the sequence, and, possibly, trimming the sequence. During our experiments, we did not observe statistically significant improvements from the application of these augmentations to the supervised training pipeline. We can add those experiments to the paper if needed.
>
> > The paper has a lot of information as a laundry list that makes it harder to find the motivation and intuition behind different design choices. I will highly recommend to restructure the paper with stronger motivation on the choice of model in fig 1.
>
> We will add motivation for the proposed method in the revised version. Here is a brief overview. Our goal was to design a self-supervised pre-training method that would be effective for the discrete event sequences domain. We designed our method with the key property of the data in mind: single events are noisy and have only a weak relation with the context. Having this property in mind, we designed CoLES to consider representative subsequences of events instead of single tokens.

---

### Official Review · AnonReviewer3 · 2020-10-30
**Well-written with extensive evaluation, but novelty seems limited**

**Rating:** 6
**Confidence:** 4

**Review:**

Summary:
- The paper proposes CoLES that uses contrastive learning to learn representations of event sequence related to user behavior (e.g., credit card transactions, retail purchase history). The method trains in a self-supervised manner by randomly slicing event sequences to generate sub-sequences. Sub-sequences from the same user event sequence are positive samples (otherwise are negative samples) for contrastive learning. The authors provide theoretical analysis for the random slicing approach, under certain assumptions of the data. They compared CoLES with other supervised, self-supervised, and semi-supervised baseslines on multiple datasets, and show the proposed method outperforms baselines.

Strong points:
- The paper is well-written and the related work provides a good context of recent advances.
- The evaluation is very thorough and extensive.
- The random slicing approach is theoretically analyzed.

Weak points:
- While the evaluation is solid, the improvements over baselines seem limited (1~2%).
- Modeling-wise the technical novelty seems limited. Using contrastive learning for self-supervision was proposed by past work (Chen et al 2020 and references therein). Apart from the analysis for using random slicing as the data augmentation scheme, limited novelty was introduced to adapt contrastive learning for discrete event sequences.
- The theory proves that random splicing generates representative samples. However, this doesn’t seem very interesting when the assumption is that the data is cyclostationary.

Recommendation:
- I’m on the borderline but slightly inclined to accept the paper. The main reason is that the paper is well-written and the evaluation is solid. The main reason for rejection would be not enough technical novelty and limited improvements.

Comments & questions:
- The paper assumes “periodicity and repeatability” in the data and bases the theoretical analysis on this assumption. However, no supporting data was shown that these event sequences do meet these assumptions. Does this assumption really make sense? Answering that will make the claim much stronger and might further explain the results across different datasets.
- How different is event sequence modeling vs language modeling? The sequence of words resembles the events. The encoder in this paper seems overly simple compared to SOTA NLP models. Would the method perform better with a better encoder design?
- What is the ratio of labeled and unlabeled data in the evaluation? This is important for comparing results with supervised approaches.
- The paper uses a lot of phrases like “significantly outperforms”, “significant increase in performance”. However, the results show 1~2% improvements which do *not* match the statements. I think the authors should revise the claims for the paper to get accepted, so that the statements reflect the results more truthfully.

Minor comments:
- In Theorem 1, the definitions of variable m & k are unclear just from the text.
- In Table 2, adding plus signs before positive results may improve readability.

==== Updates after the response ====

I thank the authors for answering my questions and the updated manuscript. I’m keeping my score and recommendation.

---

> ### Author Response · Authors · 2020-11-18
> **Answers to the first reviewer**
>
> We are very grateful to you for your attentive reading and important comments. We address your questions below:
>
> > The paper uses a lot of phrases like “significantly outperforms”, “significant increase in performance”. However, the results show 1~2% improvements which do not match the statements. I think the authors should revise the claims for the paper to get accepted, so that the statements reflect the results more truthfully.
>
> Although 1-2% improvements might seem not very significant themselves, such improvements lead to really impressive additional revenue in different industries, including financial one. Please note that our paper proves the superiority of self-supervised learning methods to a wide range of industrial cases with a very important practical domain of event sequences. This is our main finding.
>
> In Section 4, Subsection “Business applications”, we mention that in addition to the described experiments on public datasets, we have performed an extensive evaluation of our method on the private data in a large European bank. The implementation of our method brings hundreds of millions of dollars of revenue yearly in that bank alone. We observe a significant increase in performance (+ 2-10% AUROC) after the addition of CoLES embeddings to the existing models in many downstream tasks, including credit scoring, marketing campaign targeting, cold start product recommendations, fraud detection, and legal entities connections prediction.
>
> > The paper assumes “periodicity and repeatability” in the data and bases the theoretical analysis on this assumption... Does this assumption really make sense?
>
> We will clarify this assumption in a revised version. We have produced a series of plots, which show that the considered datasets have periodicity and repeatability properties. See Figure 4 in the Appendix, Section D. The plots demonstrate that the KL-divergence between random sub-sequences of the same sequence of events is relatively small compared to typical values of KL-divergence between sub-sequences of different sequences of events. This observation supports our assumption made in Section 3.2 that event sequences have a repeatable structure, and the same types of events repeat throughout the whole sequence.
>
> > How different is event sequence modeling vs language modeling? ..Would the method perform better with a better encoder design?
>
> This is an important question. Note that it was not clear in advance whether self-supervised learning is applicable to event sequence data, and our paper is the first work that discovers the superiority of this direction. In the revised version, we will explain the main differences of event sequences from other domains. In particular, texts are similar to audio and computer vision in the sense that the data of this type is "continuous": a short term in NLP can be accurately reconstructed from its context (like a pixel from its neighboring pixels). This fact underlies SOTA NLP approaches such as BERT’s Cloze task and SOTA audio approaches, like CPC. In contrast, for many types of “discrete” event sequence data, a single token cannot be determined using its nearby tokens, because the mutual information between a token and its context is small. Instead, we can model some aggregated information extracted from a segment of the event sequence, as our method does.
>
> Also note that we considered different alternatives to the architecture of the Encoder network and provided results in the Appendix, Table 6. Transformer based architecture showed less performance than RNNs.
>
> > What is the ratio of labeled and unlabeled data in the evaluation?
>
> We will add the ratio of labeled data in the description of the datasets. For “Age group” and “Churn” datasets, half of the available data is unlabelled. For the “Assessment” dataset, a major part of the data is unlabelled, only 17,7k sequences out of 330k are labeled. For the “Retail” dataset, all data is labeled.
>
> > In Theorem 1, the definitions of variable m & k are unclear just from the text.
> > In Table 2, adding plus signs before positive results may improve readability.
>
> We will clarify the definitions of variables m & k in Theorem 1 and will update
> Table 2 in the revised version.

---

### Author Response · Authors · 2020-11-24
**To all the reviewers: last important notes**

We see some general concerns about the significance of our results. To this end, we would like to emphasise important points of our paper:

- The domain of event sequences is very important in a wide range of business applications, including finance, online services, e-commerce, recommender systems, telecom, and etc. This practical domain is not similar to the domains of texts, audio, or CV (see our reply to the first reviewer).
- Before our work, there was no evidence that any self-supervised learning methods can show superiority in the discrete event sequences domain.
- We implemented our method in one European bank, and it already brings hundreds of millions of dollars yearly.

---

### Decision · Program_Chairs · 2021-01-07
**Final Decision**

**Decision:**

Reject

**Comment:**

This work is well written and accurately covers the context and recent related work. It's a good example of how to apply self-supervised training to the event sequence domain. However, the combination of a lack of technical originality (composing a set of previously explored ideas) and significant improvements in results (results with CoLES overlap in error bars with RTD results) limits the impact of this paper.

Pros:
- Well written.
- Extensive evaluation.
- Well formulated problem.

Cons:
- Lack of technical novelty. The method appears to be general to all sequences rather than specialized for event sequences so the motivation for this design is not crystal clear.
- Minor improvement in results from using the method despite written claims that the method 'significantly outperforms'.
- Limited analysis that shows the periodicity and repeatability in the data.